# The Transcript Levels and the Serum Profile of Biomarkers Associated with Clinical Endometritis Susceptibility in Buffalo Cows

**DOI:** 10.3390/vetsci11080340

**Published:** 2024-07-27

**Authors:** Ahmed El-Sayed, Salah H. Faraj, Basma H. Marghani, Fatmah A. Safhi, Mohamed Abdo, Liana Fericean, Ioan Banatean-Dunea, Cucui-Cozma Alexandru, Ahmad R. Alhimaidi, Aiman A. Ammari, Attia Eissa, Ahmed Ateya

**Affiliations:** 1Department of Animal Health and Poultry, Animal and Poultry Production Division, Desert Research Center (DRC), Cairo 11753, Egypt; decernes@drc.gov.eg; 2Department of Biology, College of Science, University of Misan, Maysan 62001, Iraq; salah81ss@uomisan.edu.iq; 3Department of Biochemistry, Physiology, and Pharmacology, Faculty of Veterinary Medicine, King Salman International University, South of Sinai 46612, Egypt; basma.hamed@ksiu.edu.eg; 4Department of Physiology, Faculty of Veterinary Medicine, Mansoura University, Mansoura 35516, Egypt; 5Department of Biology, College of Science, Princess Nourah bint Abdulrahman University, P.O. Box 84428, Riyadh 11671, Saudi Arabia; faalsafhi@pnu.edu.sa; 6Department of Animal Histology and Anatomy, School of Veterinary Medicine, Badr University in Cairo (BUC), Cairo 11829, Egypt; mohamed.abdo@vet.usc.edu.eg; 7Department of Anatomy and Embryology, Faculty of Veterinary Medicine, University of Sadat City, Sadat City 32897, Egypt; 8Department of Biology and Plant Protection, Faculty of Agricultural Sciences, University of Life Sciences King Michael I, 300645 Timisoara, Romania; ioan_banatean@usab-tm.ro; 9Tenth Department of Surgery, Victor Babeș University of Medicine and Pharmacy, 300645 Timisoara, Romania; cucui.alexandru@umft.ro; 10Zoology Department, College of Science, King Saud University, P.O. Box 2455, Riyadh 11451, Saudi Arabia; ahimaidi@ksu.edu.sa (A.R.A.); aammari@ksu.edu.sa (A.A.A.); 11Department of Animal Medicine (Internal Medicine), Faculty of Veterinary Medicine, Arish University, Arish 45511, Egypt; attia.ahmed@vet.aru.edu.eg; 12Department of Development of Animal Wealth, Faculty of Veterinary Medicine, Mansoura University, Mansoura 35516, Egypt

**Keywords:** buffaloes, gene expression, endometritis, biochemical profile

## Abstract

**Simple Summary:**

Endometritis is defined as a localized inflammatory condition of the endometrium that results in significant financial losses. This investigation used forty buffalo cows with clinical endometritis that were infected and forty seemingly healthy buffalo cows who served as the control group made up the two groups of buffalo cows. The expression levels and the serum characteristics of immune and antioxidant biomarkers linked to clinical endometritis risk varied between the investigated two categories of buffalo cows. The alteration in the profile of explored markers suggests a potential source for uterine health indicators in buffaloes.

**Abstract:**

Determining the gene expression and serum profile of the indicators linked to clinical endometritis susceptibility in Egyptian buffalo cows was the aim of this investigation. The buffalo cows that were enrolled were divided into two groups: forty infected buffalo cows with clinical endometritis and forty seemingly healthy buffalo cows that served as the control group. For the purposes of gene expression and biochemical analysis, ten milliliters of blood was obtained via jugular venipuncture from each buffalo cow. *TLR4*, *IL-8*, *IL-17*, *NFKB*, *SLCA11A1*, *NCF4*, *Keap1*, *HMOX1*, *OXSR1*, *ST1P1*, and *SERP1* were manifestly expressed at much higher levels in the buffaloes with endometritis. On the other hand, the genes that encode *SOD*, *CAT*, *NDUFS6*, *Nrf2*, and *PRDX2* were down-regulated. There was a significant (*p* < 0.05) elevation of the serum levels of non-esterified fatty acids (NEFAs), beta hydroxy butyric acid (BHBA), triglycerides (TGs), globulin, creatinine, and cortisol, along with a reduction in the serum levels of glucose, cholesterol, total protein albumin, urea, estrogen (E2), progesterone (P4), follicle-stimulating hormone (FSH), luteinizing hormone (LH), thyroxine (T4), prostaglandin F2 α (PGF2α), calcium, iron, and selenium, in the endometritis group in comparison with the control. However, no significant change was observed in the values of phosphorus, magnesium, copper, or zinc in either group. Within the selective breeding of naturally resistant animals, the variation in the genes under study and the changes in the serum profiles of the indicators under investigation may serve as a reference guide for reducing endometritis in Egyptian buffalo cows.

## 1. Introduction

Buffaloes are the primary source of premium meat and milk in Egypt and some other developing nations, even though they are typically kept in unfavorable conditions and have a restricted capacity for reproduction [1]. Endometritis is defined as a localized inflammatory condition of the endometrium that results in significant financial losses due to the treatment and milk disposal costs associated with the use of certain antibiotics, as well as due to reductions in the amount of milk produced by the animals affected [2,3]. Cows with uterine illnesses may show worsening reproductive performance, which could lead to an increase in the involuntary culling of these animals. This includes reduced pregnancy at first AI, frequencies of pregnancy, and conception rates; a delay in the restart of ovarian activity after calving and increased days open; and the prevalence of anovular cows, with increased time between calving and conception [4,5,6]. Due in large part to inadequate sanitation, the structure of their vulval lips, vaginal stimulation for milk letdown, and their wallowing habits, buffalo cows have a much higher prevalence rate of uterine infection than cows [7]. Uterine infection is also the cause of several major reproductive issues that affect buffalo cows [8,9].

According to earlier research, Trueperella pyogenes, Escherichia coli, and Fusobacterium necrophorum are the primary bacterial species linked to endometritis in dairy cows [3,6,10,11]. The incidence of uterine disorders is significantly influenced by a number of risk factors. These risk factors can be linked to uterine injury, metabolic stress, and/or inadequate cleanliness [12,13]. The risk factors for uterine infection include some that can cause endometrial trauma: stillbirth, twin, male, and beef-sired calves; dystocia; cesarean sections; and placenta retention [14]. Other factors include endocrine disorders, deficiencies in selenium, vitamin E, vitamin A, or β-carotene, the calf failing to suckle, hypocalcemia, and poor hygiene, which predispose cows to uterine diseases during the early postpartum period [15]. In fact, a number of nonantimicrobial therapeutics such as meloxicam treatment and progesterone have been studied over the past ten years, with promising outcomes for the treatment and prevention of endometritis and metritis [16]. Meanwhile, ketoprofen, a nonsteroidal anti-inflammatory drug (NSAID), is approved for use in dairy cows in Canada with a zero withdrawal time for milking. In addition, ketoprofen was recently approved for use in beef cattle and replacement dairy heifers but not for lactating cows in the US. As an NSAID, ketoprofen has analgesic, antipyretic, and antiendotoxic effects [17]. Blood biochemical analyses can be used to evaluate an animal’s general health since they provide a wealth of information about the nutritional state, overall health, and well-being of an animal [18,19]. The deviation of certain blood values from their usual ranges could be used to determine the extent of the damage to the body’s tissues [20].

The improved health of animals could be a benefit of using advanced molecular genetic approaches as an auxiliary to disease control [21]. It has been possible to identify a number of genetic biomarkers for disease resistance or susceptibility in cattle [22]. This implies that the degree of sensitivity or resistance to a disease varies throughout host genomes [23]. The biochemical, hormonal, and gene expression changes linked to endometritis in buffalo cows are still poorly understood. Thus, the current study’s objectives were to investigate the gene expression of endometritis-related biomarkers in buffalo cows and evaluate the diagnostic use of serum profiles of biochemical and hormonal markers.

## 2. Material and Methods

### 2.1. Animals and Study Design

Overall, 160 Egyptian buffalo cows were evaluated. Eighty Egyptian buffalo cows, weighing between 550 and 650 kg (mean ± SD: 600 ± 40.82) and with an average age of 7–12 years (mean ± SD: 9.42 ± 1.8), were used in this study because laboratory biochemical analysis and determining the gene expression patterns of their antioxidant and immune transcript levels were time-consuming and expensive. The experiment was conducted at the Institute for Animal Reproduction Research. The animals were kept in barns and fed grass and given water ad lib. Each buffalo cow received around 3 kg of commercial concentrate per day. The buffalo cows under investigation underwent clinical examination, which including taking their temperature, heart rate, and respiration rate [24]. The forty buffalo cows in the control group were in good health and had typical postpartum and calving conditions (regular body temperature, regular feed consumption, and no uterine discharge), while the clinical endometritis group consisted of forty buffalo cows suffering from the disease. At 28–33 DIM, a diagnosis of purulent vaginal discharge was evaluated by palpation to check their vaginal discharge. More than 21 days after calving, 40 buffalo cows were enrolled; these cows were characterized by the presence of purulent (>50% pus) uterine discharge detectable in the vagina and muco-purulent (>50% pus–50% mucus) uterine discharge detectable in the vagina more than 26 days after calving [25]. The buffalo cows were synchronized at 45–50 DIM using the Ovisynch protocol in the manner described as follows: The buffalo cows were given 100 µg of GnRH in the form of gonadorelin (Gonavet^®^, Veyx, Schwarzenborn, Germany) on day zero. On day seven, the same cows were given 500 µg of PGF2α in the form of cloprostenol (PGF Veyx Forte^®^, Veyx, Germany). Finally, on day nine, the same cows were given an injection of 100 µg of GnRH (Gonavet^®^, Veyx, Schwarzenborn, Germany).

### 2.2. Blood Sampling

At 10:00 in the morning, 28–33 days after calving, each buffalo cow underwent a jugular venipuncture to extract ten milliliters of its blood. For serum and whole blood, respectively, the drawn blood was added to plain tubes (i.e., devoid of anticoagulants) and others that included EDTA. Following cooling on crushed ice, all the samples were sent to the lab right away for additional processing. For the extraction of RNA, the tubes containing whole blood were utilized, while that in plain tubes was kept undisturbed for 15–30 min at room temperature and centrifuged at 3000 rpm for 15 min. The resulting supernatant is designated as serum. Following centrifugation, it is important to immediately transfer the liquid component (serum) into a clean microcentrifuge tube using a pipette. The samples should be maintained at 2–8 °C while being handled. If the serum is not analyzed immediately, it should be stored and transported at −20 °C or lower for subsequent biochemical analyses of energetic and oxidative stress markers. It is important to avoid multiple freeze–thaw cycles because this is detrimental to many serum components. Samples that are hemolyzed, icteric, or lipemic can invalidate certain tests.

### 2.3. Transcript Levels of Immune and Antioxidant Genes

All of the RNA from the blood samples obtained from the buffaloes under examination was extracted using Trizol solution (RNeasy Mini Ki, 74104, Product No.), in accordance with the manufacturer’s instructions. The amount of RNA that was isolated was measured and verified through the use of a NanoDrop^®^ ND-1000 spectrophotometer. The producer’s method was used to make the complementary nucleic acid for each sample (Waltham, MA, USA: Thermo Fisher, Product No. EP0441). Using SYBR Green PCR Master Mix and quantitative PCR (2× SensiFastTM SYBR, Bio-line, CAT No. Bio-98002, London, UK), the expression profiles of the immunological (*TLR4*, *IL-8*, *IL-17*, *NFKB*, *SLCA11A1*, and *NCF4*) and antioxidant (*SOD*, *CAT*, *NDUFS6*, *Nrf2*, *Keap1*, *PRDX2*, *HMOX1*, *OXSR1*, *ST1P1*, and *SERP1*) genes were assessed. Each sample’s relative amount of mRNA was measured using the Quantitect SYBR Green PCR reagent (Toronto, ON, Canada; Catalogue No. 204141).

Primers with sense and antisense sequences were created using the PubMed genome of *Bubalus bubalis* (Table 1). The *ß-actin* gene served as the constitutive normalization reference. In all, 3 µL of total RNA, 4 µL of 5× Trans Amp buffer, 0.25 µL reverse transcriptase, 0.25 µL of each primer, 12.5 µL 2× Quantitect SYBR green PCR master mix, and 4.75 µL RNase free water made up the 25 µL reaction mixture. The completed reaction mixture was then subjected to the subsequent processes in a heater cycler: use of the primer binding temperatures as indicated in Table 1 with a 1 min extension at 72 °C, preliminary denaturation for 8 min at 95 °C, reverse transcription for 30 min at 55 °C, and 40 cycles of 15 s at 95 °C. A melting curve study was performed after the amplified product was amplified to demonstrate its specificity. The differences in each gene’s expression were investigated using the 2^−ΔΔCt^ method, which compared the mRNA level of each marker in the test sample to that of the *ß-actin* gene [26,27].

### 2.4. Biochemical Analysis

Commercial kits were used according to the standard protocol of the suppliers to quantify each of the following: serum non-esterified fatty acid (NEFA), luteinizing hormone (LH), cortisol, follicle-stimulating hormone (FSH), progesterone, estradiol, albumen, thyroxine (T4), beta hydroxy butyric acid, prostaglandin F2α (PGF2α), total protein, calcium, creatinine, glucose, and triglycerides (MyBioSource ELISA kits, Southern California, San Diego, CA, USA; catalog nos. MBS748204, MBS700951, MBS701325, MBS705623, MBS704979, MBS700251, MBS9310864, MBS702370, MBS705232, MBS027214, MBS9310577, MBS754453, MBS745220, MBS7220981, and MBS031328, respectively). Cholesterol (Gamma Trade Company, Giza, Egypt); phosphorus, magnesium, and selenium (Bio-Diagnostic, Giza, Egypt); and urea (Spectrum Company, Cairo, Egypt) were measured on a selective chemistry analyzer (Apple 302, Cupertino, CA, USA), and globulin was calculated by subtracting the albumin values from the total serum protein values. The kit for Cu was from SIGMA-ALDRICH Co., Saint Louis, MO, USA, while that for Zn was from Abnova Co., Taipei City, Taiwan, and that for iron (Fe) was from Abcam Co., Cambridge, UK.

### 2.5. Statistical Analysis

A statistical software package (SPSS, ver. 20, Inc., Chicago, IL, USA) was used to perform the statistical analyses. We ran descriptive statistics for every parameter. To examine the data, Student’s *t*-test was employed. At *p ˂* 0.05, the results were deemed statistically significant. The sample size used in this study was determined using the sample size determination formula as follows:$$\mathrm{Sample}\left(N\right)=\frac{{\left(Z1-\alpha /2\right)}^{2}P\left(1-P\right)}{d^{2}}$$
where Z1 − α/2 = the standard normal variant at 5% type I error (*p* < 0.05); P = expected prevalence based on a previous study [28]; and d = absolute error or precision (which is 5%). The two groups of buffalo cows that were examined—the healthy and the endometritis buffalo cows—were the fixed effects in our model. The biochemical parameters (serum levels of glucose, cholesterol, triglycerides, globulin, creatinine, NEFA, BHBA, total protein albumin, urea, estrogen, progesterone, FSH, LH, T4, PGF2α, calcium, iron, selenium, cortisol, phosphorus, magnesium, copper, and zinc) and the immunological and antioxidant transcript levels (TLR4, IL-8, IL-17, NFKB, SLCA11A1, NCF4, Keap1, HMOX1, OXSR1, ST1P1, and SERP1, SOD, CAT, NDUFS6, Nrf2, and PRDX2) in the healthy buffalo cows and those with endometritis were the random effects in our model.

## 3. Results

### 3.1. Patterns of Immune and Antioxidant Marker mRNA Levels

The immunological and antioxidant transcript levels in healthy buffaloes and buffaloes with endometritis are displayed in Figure 1 and Figure 2, respectively. The buffaloes with endometritis had considerably higher expression levels of *TLR4*, *IL-8*, *IL-17*, *NFKB*, *SLCA11A1*, *NCF4*, *Keap1*, *HMOX1*, *OXSR1*, *ST1P1*, and *SERP1*. Conversely, there was a down-regulation of the genes that encode *SOD*, *CAT*, *NDUFS6*, *Nrf2*, and *PRDX2*. For the buffaloes with endometritis, *TLR4* had the highest possible quantity of mRNA (2.62 ± 0.16) while *CAT* had the lowest amount (0.52 ± 0.13 mRNA) for each gene. Out of all the genes examined in the healthy buffaloes, *SOD* had the highest potential level of mRNA (2.46 ± 0.14), while *OXSR1* had the lowest amount (0.32 ± 0.09).

### 3.2. The Biochemical Profile

Biochemically, the present study showed significantly (*p* < 0.05) low values for the serum levels of glucose, cholesterol, total protein albumin, urea, estrogen, progesterone, FSH, LH, T4, PGF2α, calcium, iron, and selenium (46 ± 0.5 mg/dL, 77.6 ± 9.2 mg/dL, 4.5 ± 0.2 g/dL, 3.3 ± 0.05 gm/dL, 56 ±.1.1 mg/dL, 69 ± 3.5 pg/mL, 0.5 ± 0.04 ng/mL, 3.7 ± 0.01 mU/mL, 2.7 ± 0.1 mU/mL, 4.7 ± 0.3 ng/mL, 34 ± 1.1 pg/mL, 5.4 ± 0.1 mg/dL, 115 ± 0.5 Ug/dL, and 2.8 ± 0.05 Ug/dL, respectively) in the endometritis group compared with the healthy buffaloes. Conversely, there was a noteworthy (*p* ˂ 0.05) rise in the serum concentrations of triglycerides, globulin, creatinine, NEFA, BHBA, and cortisol (67 ± 4.6 mg/dL, 0.7 ± 0.05 g/dL, 0. 5 ± 0.02 mg/dL, 0.3 ± 0.008 mmol/L, 1.4 ± 0.05 mmol/L, and 27 ± 1.1 ng/mL, respectively) in the buffalo cows with endometritis compared to the healthy group. However, no significant change was observed in the values of phosphorus, magnesium, copper, or zinc in either group (Table 2).

## 4. Discussion

The purpose of the study was to prove that the serum profiles of metabolic and hormonal markers and gene expression patterns may be utilized as diagnostic standards for clinical endometritis in Egyptian buffalo cows. According to [9], one of the most important diseases that affects cows is postpartum uterine infection. Because of its negative effects on reproductive function, which include decreased chances of conception, higher numbers of services required for each conception, a longer time between calving and first service, and decreased rates of pregnancy, there are considerable financial losses [37].

An intriguing aspect of this experiment was analyzing the mRNA expression of immunological, metabolic, and antioxidant genes in both normal buffaloes and those with clinical endometritis. We suggested that genetic variation in the buffaloes’ transcriptional response to the development of the disorder may have an impact on the course of postpartum endometritis. The following genes were assessed for their expression levels using real-time PCR in both the normal and endometritis-affected buffaloes: immune (*TLR4*, *IL-8*, *IL-17*, *NFKB*, *SLCA11A1*, and *NCF4*) and antioxidant (*SOD*, *CAT*, *NDUFS6*, *Nrf2*, *Keap1*, *PRDX2*, *HMOX1*, *OXSR1*, *ST1P1*, and *SERP1*). Our findings showed that the buffaloes with endometritis had significantly higher *TLR4*, *IL-8*, *IL-17*, *NFKB*, *SLCA11A1*, *NCF4*, *Keap1*, *HMOX1*, *OXSR1*, *ST1P1*, and *SERP1* expression levels. On the other hand, the genes encoding *PRDX2*, *NDUFS6*, *Nrf2*, *SOD*, and *CAT* were down-regulated. This study looks at these indicators’ mRNA levels and how they relate to the prevalence of postpartum endometritis in buffaloes for the first time. To examine the processes controlling the investigated gene regulations in both normal and endometritis-affected buffaloes, we employed gene expression.

Regarding the gene expression profiles of immunological and antioxidant markers in the buffaloes, the expression levels of the following genes were significantly higher in the buffaloes affected by endometritis than in the resistant ones: *A2M*, *ADAMTS20*, *KCNT2*, *MAP3K4*, *MAPK14*, *FKBP5*, *FCAMR*, *TLR2*, *IRAK3*, *CCl2*, *EPHA4*, and *iNOS*. In the buffaloes affected by endometritis, the expression levels of the *RXFP1*, *NDUFS5*, *TGF-β*, *SOD3*, *CAT*, and *GPX* genes were significantly reduced [38]. In the buffaloes with endometritis, the cytokine gene expression in the uteri of *Bubalus bubalis* associated with endometritis infection was shown to be higher than that in the control animals. Conversely, *TNF-α* and *IL-10* mRNA expression was 0.4- and 0.2-fold lower, respectively, than in the infected buffaloes [39]. According to [40], buffaloes with postparturient endometritis exhibited considerably higher expression levels of the immunological genes *IKBKG*, *LGALS*, *IL1B*, *CCL2*, *RANTES, MASP2*, *HMGB1*, and *S-LZ*.

Compared to resistant cows, endometritis-affected cows produced considerably more of the genes *TLR4*, *LITAF*, *TNF-α*, *TKT*, *RPIA*, *TLR7*, *TNF-α*, *TKT*, and *AMPD1*. Meanwhile, the expression of the IL10, ATOX1, and GST genes significantly decreased [41]. According to [42], postpartum sheep showed noticeably higher amounts of mRNA for *IL1-ß*, TNF alpha, *IL5*, *IL6*, *TLR4*, and *Tollip*. On the other hand, their SOD and CAT gene levels were much reduced. The levels of the *PRLR*, *LTF*, *CLA-DRB3.2*, beta defensin, *TLR2*, *TLR4,* and *CCL5* genes were significantly up-regulated in postparturient goats with endometritis compared to tolerant ones, while the *GPX4*, *GST*, *SOD3*, *CAT*, and *ATOX1* gene patterns demonstrated the opposite tendency [43].

Previous reports have linked the examined immunological and antioxidant markers to economically significant infectious illnesses in cattle. For example, the mRNA levels of the *NFkB* gene were significantly greater in Holstein and Montbéliarde dairy cows with mastitis than in healthy cows [44]. In comparison to healthy dairy cows, mastitic Holstein and Brown Swiss dairy cows had significantly lower levels of *SOD1*, *CAT*, *GPX1*, and *AhpC/TSA* gene expression [45]. The expression of the TLR4, OXSR1, SERP2, and ST1P1 genes was substantially higher in mastitic camels (*p* < 0.05). The *CAT*, *SOD3*, *PRDX6*, and *NDUFS6* genes, on the other hand, produced a different pattern [46]. The genes *CD-14*, *CCL2*, β defensin, *SPP1*, *BP1*, *A2M*, *TLR7*, and *TLR8* are significantly more expressed in mastitis-affected goats than in resistant goats; on the other hand, the genes *ATP1A1*, *SOD1*, *CAT*, *AhpC/TSA*, *PRDX2*, *PRDX4*, *NQO1*, and *Nrf2* are significantly less expressed in mastitis-affected goats than in resistant goats [47]. Pneumonic goats had much higher levels of *SLC11A1* gene mRNA than healthy goats did [48]. Compared to resistant Holstein dairy calves, diarrheal calves had significantly higher expression levels of *Keap1* and *HMOX1*. On the other hand, a different pattern formed for the *Nrf2* and *PRDX2* genes [49].

TLRs trigger both innate and acquired immune responses and improve neutrophil recruitment by controlling the synthesis of various chemokines and pro-inflammatory cytokines [50]. Pattern recognition receptor (PRR) SNPs can influence how the body reacts to an infection, as well as an individual’s capacity to ward off disease or build resistance to it [51]. TLR4 recognizes lipopolysaccharides (LPSs) from Gram-negative bacteria like Escherichia coli [52]. Cytokines and NF-κB function as indirect markers in inflammatory conditions [53]. NF-κB plays a role in the control of inflammasomes and stimulates the expression of certain pro-inflammatory genes, such as those that code for cytokines and chemokines. Furthermore, NF-κB is essential for controlling the proliferation, differentiation, and survival of inflammatory T cells and innate immune cells. As such, the dysregulated activation of NF-κB plays a role in the pathogenic mechanisms of many inflammatory diseases [54]. One of the most well-known likely candidate genes for innate immunity against a variety of intracellular pathogens is the trans-membrane protein *SLC11A1* [55]. Cattle mastitis appears to be related to and influenced by the innate immunity gene neutrophil cytosolic factor 4 (NCF4) [56,57].

By preventing reactive oxygen species (ROS) from entering the environment, limiting their synthesis, or protecting transition metals, which are required to create ROS, antioxidants provide defense [58]. The body’s own enzymatic and non-enzymatic antioxidant defenses, such as catalase (CAT) and superoxide dismutase (SOD), are examples of endogenous antioxidant markers that are engaged in these processes [58]. The NADH:ubiquinone oxidoreductase subunit S6 (*NDUFS6*) gene encodes the first enzyme complex in the mitochondrial electron transport chain, known as NADH:ubiquinone oxidoreductase (complex I) [59]. Electrons from NADH are transferred to the respiratory chain by this complex. This gene is altered, resulting in mitochondrial complex I impairment, which can afflict neonates and adults with neurological diseases [59]. The BTA20 region of the genome, where the *NDUFS6* gene is found in cattle, has a quantitative trait locus for somatic cell score (SCS) [60,61].

The main inducible defense against oxidative stress is the Keap1-Nrf2 stress response system, which controls the expression of cytoprotective genes [62]. Under typical circumstances, Keap1 serves as a substrate adaptor for the cullin-based E3 ubiquitin ligase, which ubiquitinates and degrades Nrf2 to prevent its transcriptional activity [63]. This could account for the different expression patterns of the *Keap1* and *Nrf2* genes shown during our study. Because of a conserved ionized thiol, the peroxiredoxin (PRDX) family of antioxidant enzyme oxidoreductase proteins can catalyze hydrogen peroxide (H2O2). Thiol-specific peroxidase serves as a sensor for signaling events brought on by hydrogen peroxide and supports cell defense against oxidative stress by detoxifying peroxides and radicals containing sulfur [64].

In the heme catabolic pathway, heme oxygenase (HMOX) is the enzyme that limits the rate at which heme is broken down into equimolar amounts of free iron, biliverdin, and carbon monoxide (CO) [65]. It is also known to be a stress-responsive protein, and because of its anti-inflammatory, anti-apoptotic, anti-coagulation, anti-proliferative, and vasodilator properties, it is believed to play a variety of protective roles against various stresses [66].

The oxidative stress-responsive kinase 1 (*OXSR1*) gene encodes the serine/threonine protein kinase (OSR1), which controls downstream kinases in response to environmental stressors [67]. The *OXSR1* expression profile throughout the periparturient period was significantly up-regulated in dromedary camels at (−14) and (+14) compared to parturition, when the lowest levels occurred [68]. The accumulation of unfolded proteins in the endoplasmic reticulum (ER stress) is connected with protein-coding genes known as stress-associated endoplasmic reticulum proteins (SERPs). It is possible that SERPs assist in proper glycosylation and prevent unfolded target protein degradation [69]. HSP70 and HSP90′s functions in protein folding are regulated and coordinated by the adaptor protein stress-induced phosphoprotein (STIP1) [70]. Furthermore, *STIP1* is expressed in response to cellular physiological stresses caused by many factors, such as high temperatures [71].

In dairy cattle, multi-pathogen bacterial infections of the vaginal tract occur after urination [72]. An inflammatory response is triggered by the release of chemokines and cytokines due to an endometrial bacterial infection. Complement fragments and inflammatory cytokines have been shown to interfere with leucocyte recruitment during inflammation [73]. Unchecked, protracted inflammation associated with tissue damage is another feature of endometritis. This inflammation is linked to the release of molecular forms that accompany injury, worsening it further and ensuring its persistence [38]. Then, excessive ROS70 accumulation results in oxidative stress. The increased expression of molecules involved in tissue remodeling, acute-phase response, and LPS signaling is also linked to these alterations [74]. The previously described factors may be responsible for the notable change in the expression patterns of the antioxidant (*SOD*, *CAT*, *NDUFS6*, *Nrf2*, *Keap1*, *PRDX2*, *HMOX1*, *OXSR1*, *ST1P1*, and *SERP1*) and immunological (*TLR4*, *IL-8*, *IL-17*, *NFKB*, *SLCA11A1*, and *NCF4*) indicators in the buffaloes with endometritis. Therefore, we believe that bovine endometritis in the buffaloes in this study had an infectious etiology. Our real-time PCR data demonstrated a significant inflammatory response in the buffaloes affected by endometritis. Normal gene expression governs most physiological mechanisms, while the disruption of gene expression can be utilized to characterize typical pathological processes [75,76].

The innate immune response is disrupted by poor metabolic status and negative energy balance (NEB) during the postpartum period, which, in turn, contributes to the development of clinical endometritis. The current study’s findings demonstrated the value of glucose, BHBA, and NEFA as biomarkers for endometritis identification. In the current investigation, the buffaloes with clinical endometritis had significantly lower blood glucose levels and higher serum values of NEFA and BHBA. This result was in line with earlier research [36,77]. Previous research has shown a negative correlation between prepartum blood glucose concentrations and the incidence of clinical endometritis [77] and persistent postpartum bacterial infection [78].

A lack of energy, particularly hypoglycemia, causes the blood levels of BHBA and NEFA to rise. This can lead to hazardous levels and impair immunological responses, which raises the possibility of developing clinical endometritis [79]. Hypoglycemia, as seen in transition cows, reduces immune cell function and raises the risk of infection [80]. Immune cell function (proliferation and differentiation) is glucose-dependent [81]. Maternal hypoglycemia is caused by the reduced consumption of dry matter and blood glucose being partitioned towards the gravid uterus during late pregnancy and the mammary glands during lactation [82].

In contrast to healthy buffaloes, the endometritis-affected buffalo cows in the current study had markedly higher serum TG contents and much lower cholesterol levels. Ref. [83] reported similar results. Our hypothesis was that with a postpartum NEB status, body fat will have continuously been used as an energy source, leading to TG buildup in the hepatocytes and impairing liver function [84]. Since the majority of the cholesterol in the endometritis group will have come from the cows’ intestines, a decrease in feed intake may have been the cause of the group’s declining cholesterol levels [85]. As part of lipoproteins, which make up the lipid makeup of cell membranes, cholesterol is regarded as a negative acute-phase reactant. Measuring cholesterol aids in determining how well the liver functions [86]. All of these results amply demonstrated how metabolic imbalance and NEB both enhance the risk of endometritis, and vice versa.

The current study found that in the buffalo cows with endometritis, their metabolic indices, such as total protein and albumen, dramatically dropped, while globulin declined. Our findings concurred with those of previous studies [36,87,88] but differed from the findings reported by [77], who showed that the serum total protein levels were higher in cows with uterine infections than in healthy ones. The drop in the serum albumin concentrations in the diseased animals was attributed to a corresponding decrease in their serum total protein concentrations. As a negative acute-phase protein, serum albumin’s concentration falls during acute inflammations. When compared to healthy buffaloes, endometritis-affected buffaloes have significantly lower serum albumin concentrations because endometritis is an acute inflammation of the endometrium. This hypothesis is consistent with [89,90]’s findings. Due to an increase in immunoglobulins in the bloodstream as a result of the illness, the affected buffaloes’ serum globulin levels were significantly increased [88].

A significant risk factor for chronic postpartum bacterial infection has also been linked to lower serum urea nitrogen, similar to glucose [91,92]. In the current study, the endometritis-affected buffaloes showed a notable rise in their creatinine levels and a significant decline in their serum blood urea nitrogen content when compared to the healthy buffaloes. Comparable outcomes were reported by [36]. Furthermore, [93] discovered a negative relationship between urea concentrations and the expression of genes related to innate immunity and inflammation in cow uteri. On the other hand, [12] found no variations in urea concentrations between normal and clinical endometritis groups. According to [94], creatinine can be changed by the kidneys without reabsorption and represents mobilization of the skeletal muscle.

The endocrinological profiles showed that compared to the normal, unaffected animals, the endometriotic animals had considerably higher cortisol levels and decreased levels of FSH, LH, E2, P4, PGF2α, and T4. Our findings concurred with previous studies [32,95]. For ovarian cycles to resume after giving birth, normal pituitary and hypothalamus activity is essential. The follicle-stimulating hormone (FSH) concentrations in animals with uterine infections are unaltered, and ovarian follicular waves occur in the initial weeks following parturition [96]. However, in endometriotic or metritis-affected cows, it has been reported that endotoxins derived from E. coli suppress the release of GnRH and LH from the hypothalamus and the pituitary gland, respectively, as well as the pituitary gland’s sensitivity to gonadotrophin-releasing hormone. This reduces the ability of the dominant follicle to ovulate [97]. Moreover, endotoxins suppress the pituitary’s sensitivity to GnRH [98], which may have an impact on luteal development and ovulation. It has been reported that postpartum ovarian follicular growth and function are disrupted due to the suppression of pituitary LH secretion caused by bacterial loads in the uterus, bacterial metabolic products, and the concomitant inflammation of the uterine layers [99]. Reduced LH pulse frequencies in dairy cattle are mostly caused by metabolic stress, which is typically caused by a negative energy balance [93]. Bacterial contamination of the postpartum uterus has also been linked to decreased plasma P4 concentrations, in addition to smaller CLs in the first postpartum estrous cycle [100]. Ovarian activity is adversely affected by uterine infections. When compared to healthy cows, the first postpartum dominant follicle in cows with significant bacterial uterine contamination was smaller and secreted less estrogen [100]. In comparison to healthy cows, these cows also had smaller CLs and lower plasma P4 concentrations [100]. Furthermore, because the endometrial epithelial cells of animals affected by uterine disease secrete PGE2 instead of PGF2α, luteolysis is disrupted due to the change from PGF2α (luteolytic) to PGE2 (anti-inflammatory). This leads to infertility and longer luteal phases. According to reports, diseases related to uterine function are typically linked to disrupted steroidogenesis. This is because the impaired endometrium fails to produce the appropriate amount of PGF2α, which is necessary to control CL function and ultimately synthesize progesterone [101].

According to [102], cortisol may also hinder follicular growth and ovulation in ruminants, perhaps exacerbating the animals’ deficiency in reproduction. Mainly, cortisol suppresses the immune system [103]. Increased cortisol levels may be a factor in the overall greater incidence of endometritis in cows. A notable difference was noted between healthy and meteoritic-damaged cows [104], but plasma estradiol peaks immediately after calving and rapidly declines after calving [105]. The endometriotic-affected cows ultimately had altered peripheral plasma concentrations of FSH, LH, E2, P4, and prostaglandin metabolites.

When comparing the serum calcium concentrations of the endometriotic buffalo cows to those of healthy ones, we observed a drop, but we saw no effect on the concentrations of magnesium or phosphorus. Previous studies have shown that cows with endometritis, either clinical or subclinical, had lower serum calcium levels than cows without uterine illnesses [36,83,85]. In relation to reduced circulating neutrophil counts, a diminished potential for neutrophil oxidative burst, complement activation, and a heightened risk of uterine infection [106], a decrease in calcium concentrations may be linked to immunological function impairment [107]. Uterine involution is delayed by subclinical hypocalcemia, which affects myometrial contraction [108]. Magnesium is a crucial mineral for maintaining calcium homeostasis; hypocalcemia can result from a drop in magnesium content [109]. Magnesium also plays a role in opsonization processes [110]. However, there have been conflicting results regarding the involvement of magnesium in the development of clinical endometritis [111]. According to [107], cows with clinical endometritis had lower calcium concentrations than cows without the condition but had higher magnesium concentrations. Conversely, a study that did not distinguish between cows with subclinical and clinical endometritis found that the magnesium levels were lower in endometritis-affected cows than in normal ones, but calcium concentrations were not linked to endometritis [112]. On the other hand, there are not many reports on how phosphorus levels change over the course of clinical endometritis development.

This study showed that compared to healthy animals, the endometritis-affected animals had lower serum concentrations of iron and selenium. Indeed, [101] reported similar outcomes. The lower blood selenium levels in the endometritis-affected buffalo cows may have been caused by oxidative stress, whereas their lower serum iron levels may have been related to regenerative anemia.

## 5. Conclusions

Our results emphasize the importance of immunological (TLR4, IL-8, IL-17, NFKB, SLCA11A1, and NCF4) and antioxidant (TLR4, IL-8, IL-17, NFKB, SLCA11A1, and NCF4) gene diversity as a surrogate indicator for the disease under study in Egyptian buffalo cows. The findings also present strong evidence of significant biochemical and hormonal changes linked to endometritis in Egyptian buffalo cows, specifically in relation to blood sugar, cholesterol, total protein albumin, urea, estrogen, progesterone, FSH, LH, T4, PGF2α, calcium, iron, and selenium. The fluctuating patterns of expression in buffalo cows that are resistant or not resistant to endometritis may serve as biomarkers and references for monitoring the health of these animals. These findings present a viable way to reduce endometritis in Egyptian buffalo cows by selectively breeding animals according to genetic markers linked to innate immunity to infection.

## Figures and Tables

**Figure 1 vetsci-11-00340-f001:**
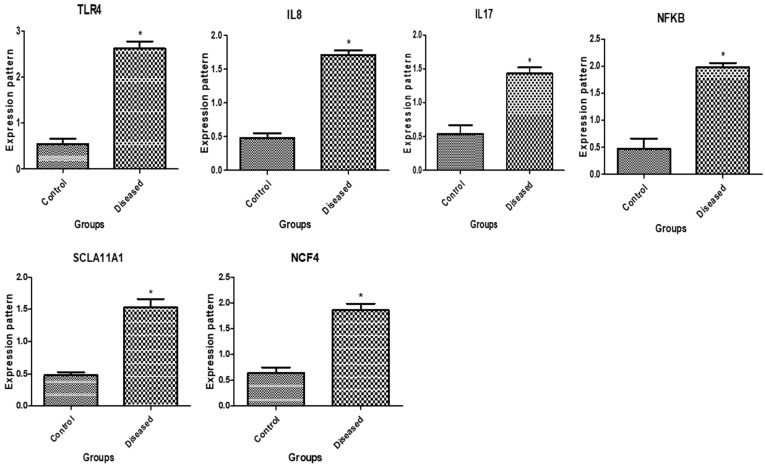
Immune gene transcript levels in normal buffaloes and those with endometritis. The asterisk (*) denotes importance at *p* < 0.05.

**Figure 2 vetsci-11-00340-f002:**
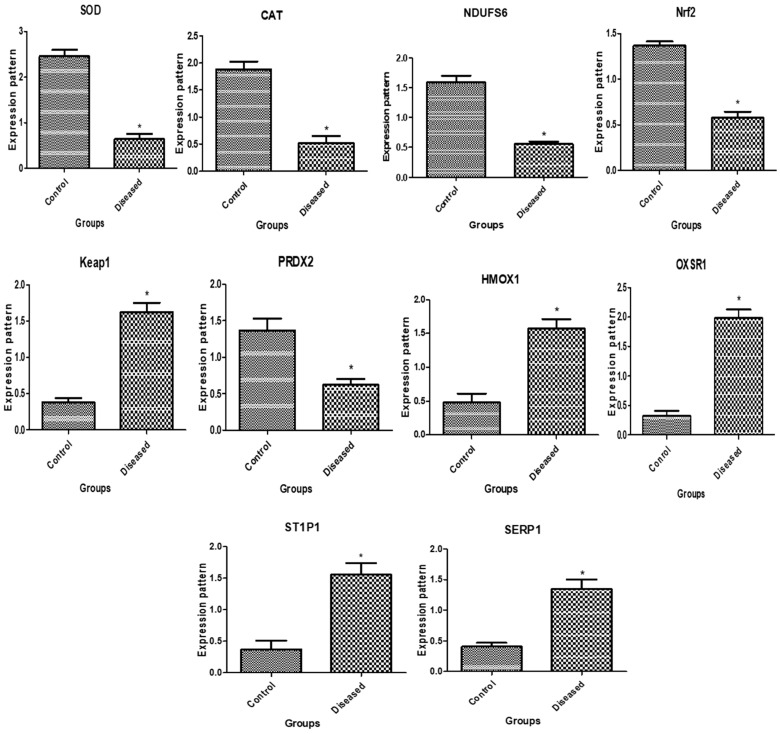
Antioxidant gene transcript levels in normal buffaloes and those with endometritis. The asterisk (*) denotes importance when *p* < 0.05.

**Table 1 vetsci-11-00340-t001:** Real-time forward and reverse PCR primers made of oligonucleotides for immune, metabolic, and antioxidant genes under study.

Investigated Marker	Primer	Product Size (bp)	Annealing Temperature (°C)	GenBank Isolate	Origin
*TLR4*	F5′-CCTGCATTGAAGCTCAGTTCTC-3 R5′-GGTTTTCTAGTTGATTTCCGCC-3′	244	58	MT424002.1	Present Research
*IL-8*	F5′-TCTCTGCAGCTCTGTGTGAA-3 R5′-GGGTGGAAAGGTGTGGAATGT-3′	94	60	NM_001290920.1	
*IL-17*	F5′-GGACTCTCCACCGCAATGAG-3′ R5′-CCTAAGCCAAATGGCGGACA-3′	249	58	OQ730437.1	
*NFKB*	F5′-CGAAAGCGAATCTCTCCTGGT-3′ R5′-TGACTGGGCCTAAGGAATGG-3′	182	58	XM_006046119.4	
*SLC11A1*	F5′-TCATGTCAGGTGACACAGGC-3′ R5′-CCAGCCTGAAGATCCGACTC-3′	247	58	XM_006046401.3	
*NCF4*	F5′-TCAGCCAACATCGCTGACAT-3′ R5′-TCCAGCTTGCTCTGTAAGGC-3′	143	60	XM_006056976.4	
*SOD*	F5′-GTCCCAGGTGCTCGACTCT-3′ R5′-ATCTCCTGCCAGATCTCCGT-3′	160	60	XM_006041479.4	
*CAT*	F5′-CTGAGTGGCGGAGTCTGAAG-3′ R5′-CTGGATTACCGCCTCCAGTG-3′	200	60	XM_044929272.2	
*NDUFS6*	F5′-GGGAGTCGGGTGATATCGTG-3′ R5′-GTCCCCGTCTTCGTTTCCTT-3′	92	60	XM_006051698.4	
*Nrf2*	F5′-GTCAGGGAGAAGCGAGTTCC-3′ R5′-TACCTCTCGACTTACCCCGA-3′	241	60	XM_006051425.4	
*Keap1*	F5′-AATCACGACTTCTTCCCCGC-3′ R5′-CTCCCGCCTAACTTTCGCTA-3′	196	60	XM_006068351.4	
*PRDX2*	F5′-ATGAGCATGGGGAAGTCTGC-3′ R5′-GAGCAGGTCTGGCATTTCCT-3′	193	60	XM_006041572.4	
*HMOX1*	F5′-CAAGCGCTATGTTCAGCGAC-3′ R5′-GCTTGAACTTGGTGGCACTG-3′	206	58	XM_045165381.1	
*OXSR1*	F5′-GATGAGCTGTGGCTCGTCAT-3′ R5′-GTGGTTGGTGTTAGCAAGGC-3′	125	60	XM_025272781.3	
*ST1P1*	F5′-AGCTGGAGCCAACCTTCATC-3′ R5′-CATCATGCAGCGCTGGTAAC-3′	155	58	XM_006050858.3	
*SERP1*	F5′-TATGGCCAACGAGAAGCACA-3′ R5′-GGTCCTACAGACGCCTTCTC-3′	147	58	XM_006048532.4	
*ß. actin*	F5′-GGAATCCTGCGGTATTCACGA-3′ R5′-CCGCCAATCCACACAGAGTA-3′	222	58	NM_001290932.1	

*TLR4* = Toll-like receptor 4; *IL-8* = interleukin-8; *IL-17* = interleukin-17; *NFKB* = nuclear factor kappa B; *SLC11A1* = solute carrier family 11 member 1; *NCF4*; neutrophil cytosolic factor 4; *SOD* = superoxide dismutase; *CAT* = catalase; *NDUFS5* = NADH: ubiquinone oxidoreductase subunit S5; *Nrf2 =* nuclear factor erythroid 2-related factor 2; *Keap1* = Kelch-like ECH-associated protein 1; *PRDX2*= peroxiredoxin 2; *HMOX1 =* heme oxygenase-1; *OXSR1* = oxidative stress-responsive kinase 1; *ST1P1* = stress-induced phosphoprotein 1; *SERP1 =* stress-associated endoplasmic reticulum protein 1.

**Table 2 vetsci-11-00340-t002:** Metabolic and hormonal biomarkers (mean ± SE) in control and buffaloes with endometritis.

Parameters	Unit	Control	Endometritis	*p*-Value	Reference Interval
Glucose	(mg/dL)	57.3 ± 0.8	46 ± 0.5	0.004	22.3–97.4 [29]
NEFA	(mmol/L)	0.1 ± 0.01	0.3 ± 0.008	0.001	0.2–0.29 [30]
BHBA	(mmol/L)	0.5 ± 0.05	1.4 ± 0.05	0. 001	0.35–0.67 [30]
Cholesterol	(mg/dL)	77.6 ± 9.2	56.6 ± 3.5	0.02	80–120 [29]
Triglycerides	(mg/dL)	43 ± 2.6	67 ± 4.6	0.01	10.3–59.3 [29]
Total protein	(g/dL)	5.7 ± 0.08	4.5 ± 0.2	0.005	5.4–9.3 [29]
Albumin	(g/dL)	4.4 ± 0.05	3.3 ± 0.05	0.001	2.2–4.6 [29]
Globulin	(g/dL)	0.4 ± 0.02	0.7 ± 0.05	0.004	3.9–4.1 [31]
Blood urea nitrogen	(mg/dL)	68.6 ± 0.1.4	56 ±.1.1	0.002	13.2–64.1 [29]
Creatinine	(mg/dL)	0.1 ± 0.04	0. 5 ± 0.02	0.002	1.07–2.52 [29]
Estrogen	(pg/mL)	90 ± 1.1	69 ± 3.5	0. 005	50–90 [32]
Progesterone	(ng/mL)	1 ± 0.06	0.5 ± 0.04	0. 004	0.9–1.3 [32]
FSH	(mU/mL)	5.6 ± 0.08	3.7 ± 0.01	0.001	3.5–6 [32]
LH	(mU/mL)	3.8 ± 0.06	2.7 ± 0.1	0.001	3.2–5 [32]
T4	(ng/mL)	8.8 ± 0. 5	4.7 ± 0.3	0.004	8.6–9.2 [33]
Cortisol	(ng/mL)	18 ± 0.5	27 ± 1.1	0.002	17.5–20 [34]
PGF2α	(pg/mL)	51 ± 1.5	34 ± 1.1	0.001	53.67 ± 2.4 [35]
Calcium	(mg/dL)	6.6 ± 0.08	5.4 ± 0.1	0.001	6.4–12.8 [29]
Phosphorus	(mg/dL)	3.5 ± 0.04	3.7 ± 0.05	0.06	3.2–3.9 [36]
Magnesium	(mg/dL)	2.4 ± 0.05	2.3 ± 0.02	0.3	2.20–3.93 [29]
Iron	(Ug/dL)	132.6 ± 1.4	115 ± 0.5	0.001	60.1–187.4 [29]
Selenium	(Ug/dL)	3.9 ± 0.08	2.8 ± 0.05	0.001	3.6–4.1 [33]
Copper	(Ug/dL)	84.6 ± 1.4	84.6 ± 2	1	51–151.1 [29]
Zinc	(Ug/dL)	67 ± 1.1	67.6 ± 0.8	0.6	52.2–130.9 [29]

## Data Availability

The appropriate author will provide the supporting information for the study’s conclusions upon reasonable request.

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
