# Peer review of "The Transcript Levels and the Serum Profile of Biomarkers Associated with Clinical Endometritis Susceptibility in Buffalo Cows"

_vetsci, 2024, doi:10.3390/vetsci11080340_

Round 1

Reviewer 1 Report

Comments and Suggestions for Authors

The authors evaluated the biomarkers profile of cows with and without PVD, even the literature has a lot of information related with this topic, I think the manuscript has merit in describing some information related in Egyptian herds.

My main concerns are related to the Materials and Methods, which make any reading related to the results impossible before the necessary explanations and corrections are made.

52-54 Cite newest references, and describe better some of reasons for economic losses including the results for the cost of veterinary, delays conception, increases artificial inseminations etc.

https://doi.org/10.1111/rda.14301

59-72 Cite the main pathogens related with endometritis.

https://doi.org/10.1111/rda.14017

66-67 introduce some information related with uterine diseases treatment

https://doi.org/10.1590/1984-3143-AR2020-0063

67-68 Exclude this sentence, those methods are not required, and the reference is related with equine.

https://doi.org/10.1016/j.theriogenology.2017.02.005

93 Clinical endometritis instead clinic endometritic

MM: Sample size calculation is missing, how many days in milk (DIM) were the cows evaluated, insert the range or interval mean ± standard deviation and number of cows enrolled at 21 days after calving and 26 after calving. When after calving were the synchronized, for example after 45 or 50 DIM.

How many cows were evaluated to meet 40 per group? How did you select the 40 control (healthy) cows?

How did you check the vaginal discharge? Metricheck, palpation?

As you did not check the PMN count, replace endometritis for purulent vaginal discharge

https://doi.org/10.1530/REP-22-0452

What were your random and fixed effects in your model?

When were the blood samples collected?

Comments on the Quality of English Language

An English review carried out by a native speaker could improve the quality of English.

Author Response

Reviewer 1

The authors evaluated the biomarkers profile of cows with and without PVD, even the literature has a lot of information related with this topic, I think the manuscript has merit in describing some information related in Egyptian herds. My main concerns are related to the Materials and Methods, which make any reading related to the results impossible before the necessary explanations and corrections are made.

We would like to express our deep thanks and appreciation to the reviewer for the time given to revise our submission.

  • Comment: 52-54 Cite newest references, and describe better some of reasons for economic losses including the results for the cost of veterinary, delays conception, increases artificial inseminations etc. https://doi.org/10.1111/rda.14301.

Response: The reviewer's informative comments are much appreciated by the authors. We have followed the reviewer's recommendations and included the most recent references regarding the financial losses caused by clinical endometritis and how it affects buffalo cows.

  • Comment: 59-72 Cite the main pathogens related with endometritis. https://doi.org/10.1111/rda.14017

Response: The reviewer's insightful remarks are much appreciated by the writers. We have enumerated the primary infections linked to endometritis based on the reviewer's recommendations.

  • Comment: 66-67 introduce some information related with uterine diseases treatment. https://doi.org/10.1590/1984-3143-AR2020-0063

Response:  The reviewer's insightful remarks are much appreciated by the writers. We have added some material about the treatment of uterine disorders in compliance with the reviewer's recommendations.

  • Comment: 67-68 Exclude this sentence, those methods are not required, and the reference is related with equine. https://doi.org/10.1016/j.theriogenology.2017.02.005.

Response:  The writers greatly value the reviewer's perceptive observations. As per the reviewer's suggestions, we have excluded this sentence.

  • Comment: 66-67 Clinical endometritis instead clinic endometritic

Response:  The reviewer's insightful remarks are much appreciated by the writers. Corrected.

  • Comment: MM: Sample size calculation is missing, how many days in milk (DIM) were the cows evaluated, insert the range or interval mean ± standard deviation and number of cows enrolled at 21 days after calving and 26 after calving. When after calving were the synchronized, for example after 45 or 50 DIM.

Response:  The writers greatly value the reviewer's perceptive observations. The sample size used in this study was determined using the sample size determination formula as follows:

Z1-α/2 = Standard normal variant at 5% type I error (P < 0.05); P = expected prevalence based on previous study (Hassan et al., 2020); d = absolute error or precision (which is 5%).

The buffalo cow was assessed at 28–33 DIM. Forty buffalo cows were enrolled at 21 days and 26 days following calving, respectively. Following calving, the buffalo cows were synchronized at 45–50 DIM.

Hassan, M., Arshad, U., ErdoÄŸan, G. and Ahmad, N. 2020. Evaluation of haemodynamic changes of uterine arteries using Doppler ultrasonography during different stages of pregnancy in Bos indicus cows. Reprod Dom Anim; 55:1425–1433. DOI: 10.1111/rda.13793

  • Comment: How many cows were evaluated to meet 40 per group? How did you select the 40 control (healthy) cows?bised sample

Response:  The reviewer's astute observations are much valued by the authors. 160 buffalo cows were assessed, with 40 in each group due to the time-consuming and costly nature of laboratory biochemical analysis and gene expression patterns of the immune and antioxidant transcript levels. The control (healthy) buffalo cows were in good health and had normal calving and postpartum conditions (i.e., customary feed consumption, body temperature, and no uterine discharge), while the endometritis group was characterized by purulent or muco-purulent uterine discharge with offensive odor, anorexia, and depression.

  • Comment: How did you check the vaginal discharge? Metricheck, palpation?

Response:  The authors greatly appreciate the reviewer's insightful observations. We used palpation to check the vaginal discharge.

  • Comment: As you did not check the PMN count, replace endometritis for purulent vaginal discharge. https://doi.org/10.1530/REP-22-0452

Response: The writers greatly value the reviewer's insightful views. We sincerely agree with the reviewer's observation that our study's lack of a PMN count and bacterial screening of the endometrial secretions may have been a limitation that should be taken into account in subsequent research. However, in this study, the diagnosis of clinical endometritis was determined by rectal palpation, followed by rectal massage, squeezing the ureine discharge with the other gloved hand, and examining the uterine discharge's color, odor, and consistency. The study's clinical endometritis was identified as expulsed purulent (>50% pus) uterine discharge detectable in the vagina more than 21 days after calving or muco-purulent (50 % pus and 50 % mucus) uterine discharge detectable in the vagina after 26 days after calving, according to Pascottini et al. (2023).

Pascottini, O.B., et al., Genesis of clinical and subclinical endometritis in dairy cows. Reproduction, 2023. 166(2): p. R15-R24.

  • Comment: What were your random and fixed effects in your model?

Response:  The reviewer's insightful views are much appreciated by the writers. The biochemical parameters (serum levels of glucose, cholesterol, triglycerides, globulin, creatinine, NEFA, BHBA, total protein albumin, urea, estrogen, progesterone, FSH, LH, T4, PGF2α, calcium, iron, selenium, cortisol, phosphorus, magnesium, copper, and zinc) and the immunological and antioxidant transcript levels (TLR4, IL-8, IL-17, NFKB, SLCA11A1, NCF4, Keap1, HMOX1, OXSR1, ST1P1, and SERP1, SOD, CAT, NDUFS6, Nrf2, and PRDX2) in healthy and endometritis buffaloes cows were the random effects of our model. The two groups of buffalo cows that were examined—the healthy and the endometritis buffalo cows—were the fixed effects of our model.

  • Comment: When were the blood samples collected?

Response:  The authors much appreciate the reviewer's insightful observations. The blood was collected from each bufalo–cow via jugular venipuncture at 10 O’clock morning.

  • Comments on the Quality of English Language
  • An English review carried out by a native speaker

Response:  The writers have expressed their sincere gratitude for the reviewer's insightful observation. The manuscript is English edited by native speaker. The manuscript is also checked for the quality of the English language using an online QuillBot grammar checker website.    https://quillbot.com/grammar-check

Reviewer 2 Report

Comments and Suggestions for Authors

Thank you for selecting me as a reviewer of this manuscript.

Please confirm my comments. Thank you.

 The manuscript entitled "Transcript Level and Serum Profile of Biomarkers Associated with Clinical Endometritis Susceptibility in Buffalo-Cows” (Manuscript ID; vetsci-3078394) was describes the to determine the gene expression and serum profile of indicators linked to clinical endometritis susceptibility in Egyptian buffalo cows. From the results, gene expressions of TLR4, IL-8, IL-17, NFKB, SLCA11A1, NCF4, Keap1, 33 HMOX1, OXSR1, ST1P1, and SERP1 were higher in endometritis group than control group, on the other hand, SOD, CAT, NDUFS6, Nrf2, and PRDX2 35 were lower in endometritis group than in control group. In serum metabolite f actors or sex hormones, NEFA, BHBA, TG, globulin, creatinine, and cortisol were higher in endometritis group than control group, on the other hand, glucose, cholesterol, total protein. albumin, urea, E2, P4, FSH, LH, T4, PGF2α, calcium, iron, and selenium were lower in endometritis group than the control group. This manuscript is potentially interesting and provides useful information for the applied reproductive information in buffalo. However, there are serious points needing corrections in the manuscript. Please consider the suggested edits listed below.

General Comments:

 There are several serious study design flaws in this study.

1. There is no description of the method used to evaluate endometritis. Was endometritis evaluated by vaginal speculum? or evaluated by cytobrush? or other methods? This point is unclear and does not give the reader sufficient information on how endometritis is evaluated.

2. When was endometritis evaluated? Also, there is no mention of when endometritis had occurred or diagnosis was made in this experiment after calving of each buffalo. This point should be stated clearly.

3. There is no description of days in milk from the calving at the time the endometritis cow was diagnosed, and furthermore, no mention of the days in milk of the blood sampling in endometritis cows. In addition, when were the blood samples taken from the control cows after calving? Without a clear description of this point, it is unclear whether the evaluation of differences in gene expression and blood composition is looking at differences due to endometritis or differences in metabolic status after calving.

Comments on the Quality of English Language

 The manuscript entitled "Transcript Level and Serum Profile of Biomarkers Associated with Clinical Endometritis Susceptibility in Buffalo-Cows” (Manuscript ID; vetsci-3078394) was describes the to determine the gene expression and serum profile of indicators linked to clinical endometritis susceptibility in Egyptian buffalo cows. From the results, gene expressions of TLR4, IL-8, IL-17, NFKB, SLCA11A1, NCF4, Keap1, 33 HMOX1, OXSR1, ST1P1, and SERP1 were higher in endometritis group than control group, on the other hand, SOD, CAT, NDUFS6, Nrf2, and PRDX2 35 were lower in endometritis group than in control group. In serum metabolite f actors or sex hormones, NEFA, BHBA, TG, globulin, creatinine, and cortisol were higher in endometritis group than control group, on the other hand, glucose, cholesterol, total protein. albumin, urea, E2, P4, FSH, LH, T4, PGF2α, calcium, iron, and selenium were lower in endometritis group than the control group. This manuscript is potentially interesting and provides useful information for the applied reproductive information in buffalo. However, there are serious points needing corrections in the manuscript. Please consider the suggested edits listed below.

General Comments:

 There are several serious study design flaws in this study.

1. There is no description of the method used to evaluate endometritis. Was endometritis evaluated by vaginal speculum? or evaluated by cytobrush? or other methods? This point is unclear and does not give the reader sufficient information on how endometritis is evaluated.

2. When was endometritis evaluated? Also, there is no mention of when endometritis had occurred or diagnosis was made in this experiment after calving of each buffalo. This point should be stated clearly.

3. There is no description of days in milk from the calving at the time the endometritis cow was diagnosed, and furthermore, no mention of the days in milk of the blood sampling in endometritis cows. In addition, when were the blood samples taken from the control cows after calving? Without a clear description of this point, it is unclear whether the evaluation of differences in gene expression and blood composition is looking at differences due to endometritis or differences in metabolic status after calving.

Author Response

Reviewer 2

We wish to thank the reviewer from the bottom of our hearts for taking the time to edit our work.

  • Comment: There is no description of the method used to evaluate endometritis. Was endometritis evaluated by vaginal speculum? or evaluated by cytobrush? or other methods? This point is unclear and does not give the reader sufficient information on how endometritis is evaluated.

Response:  The reviewer's astute observations are much appreciated by the authors. The endometritis was assessed by rectal palpation followed by rectal massage and squeezing the utreine discharge on the other gloved hand and examining the color, odor, and consistency of the uterine discharge. According to Pascottini et al. (2023), the clinical endometritis in this study was diagnosed as expulsed purulent (>50% pus) uterine discharge detectable in the vagina more than 21 days after calving or muco-purulent (50 percent pus and 50 percent mucus) uterine discharge detectable in the vagina after 26 days after calving.

Pascottini, O.B., et al., Genesis of clinical and subclinical endometritis in dairy cows. Reproduction, 2023. 166(2): p. R15-R24.

  • Comment: When was endometritis evaluated? Also, there is no mention of when endometritis had occurred or diagnosis was made in this experiment after calving of each buffalo. This point should be stated clearly.

Response:  The writers are grateful for the reviewer's perceptive observations.

The endometritis was evaluated at 28–33 day after caving. The clinical endometritis in this study was diagnosed as expulsed purulent (>50% pus) uterine discharge detectable in the vagina more than 21 days after calving or muco-purulent (50 % pus and 50 % mucus) uterine discharge detectable in the vagina after 26 days after calving (Pascottini et al., 2023). Blood was then drawn to estimate the serum profile of biochemical and hormonal markers and investigate the gene expression of endometritis-related immunological and antioxidant biomarkers in buffalo cows.

Pascottini, O.B., et al., Genesis of clinical and subclinical endometritis in dairy cows. Reproduction, 2023. 166(2): p. R15-R24.

  • Comment: There is no description of days in milk from the calving at the time the endometritis cow was diagnosed, and furthermore, no mention of the days in milk of the blood sampling in endometritis cows. In addition, when were the blood samples taken from the control cows after calving? Without a clear description of this point, it is unclear whether the evaluation of differences in gene expression and blood composition is looking at differences due to endometritis or differences in metabolic status after calving.

Response:  The writers greatly value the reviewer's perceptive opinions. Blood was drawn and the endometritis was evaluated at 28–33 after calving.
Additionally, blood samples from the control cows were taken 28–33 days after calving. Thus, differences in gene expression and blood composition are being assessed in relation to endometritis rather than differences in metabolic status after calving.

  • Comments on the Quality of English Language
  • Response:  The writers have expressed their sincere gratitude for the reviewer's insightful observation. The manuscript is English edited by native speaker. The manuscript is also checked for the quality of the English language using an online QuillBot grammar checker website. https://quillbot.com/grammar-check

Reviewer 3 Report

Comments and Suggestions for Authors

Wszystkie komentarze są załączone w pliku.

The manuscript titled „Transcript Level and Serum Profile of Biomarkers Associated with Clinical Endometritis Susceptibility in Buffalo-Cows” focuses on endometritis in Egyptian buffalo cows. The manuscript is a very valuable collection of many scientific publications that focus on a very important topic: reproduction. A considerable amount of work has been conducted and with some minor edits and additional information should be acceptable for publication. I would appreciate the authors responding to the following comments.

1.      More information about uterine infection that should caused endometritis because in introduction is only a mention.

2.      In 2.2. Blood sampling authors write “…plain tubes were kept overnight at room temperature and centrifuged at 3000 rpm for 15 minute” can they explain more? Overnight at room temperature?

3.      Why was only one reference gene used in Real Time PCR?

4.      Why was the research material exclusively blood?

5.      Why in line 207 where the authors write about their experiment the sheep is marked as an experimental model?

6.      In Table No. 2, a separate column should be designated for the unit.

7.      I would suggest changing all charts to have one color for the bars.

Comments on the Quality of English Language

Język angielski wymaga drobnej korekty.

Author Response

Reviewer 3

We like to sincerely thank and appreciate the reviewer for taking the time to edit our contribution.

  • Comment: More information about uterine infection that should caused endometritis because in introduction is only a mention.

Response:  The reviewer's insightful views are much appreciated by the writers. We have expanded the material regarding uterine infections that should result in endometritis based on the reviewer's recommendations.

  • Comment: In 2.2. Blood sampling authors write “…plain tubes were kept overnight at room temperature and centrifuged at 3000 rpm for 15 minute” can they explain more? Overnight at room temperature?.

Response:  The writers greatly value the reviewer's perceptive opinions.
As per the reviewer's recommendations, we have expanded on the details on the blood sample.

  • Comment: Why was only one reference gene used in Real Time PCR?

Response: We thank the reviewer for this.

  • We have judged our results and we have concluded that there was no conflict results to use another constitutive genes
  • We have only two categories of samples (healthy and endometritis) judged in the same conditions; we mean there are no many treatments in our investigation to use another constitutive gene
  • We applied gene expression in blood (one site for judging gene expression i.e. gene expression is not conducted in many organs). Consequently one constitutive gene expressed continuously in blood is enough
  • We thank reviewer for this. Many published papers used one constitutive gene (beta actin) in blood and the authors get good results.
  • 1- Darwish A, Ebissy E, Ateya A, El-Sayed A. Single nucleotide polymorphisms, gene expression and serum profile of immune and antioxidant markers associated with postpartum disorders susceptibility in Barki sheep. Anim Biotechnol. 2023 Apr;34(2):327-339. doi: 10.1080/10495398.2021.1964984. Epub 2021 Aug 18. PMID: 34406916.
  • 2- Al-Sharif M, Marghani BH, Ateya A. DNA polymorphisms and expression profile of immune and antioxidant genes as biomarkers for reproductive disorders tolerance/susceptibility in Baladi goat. Anim Biotechnol. 2023 Dec;34(7):2219-2230. doi: 10.1080/10495398.2022.2082975. Epub 2022 Jun 7. PMID: 35671246.
  • 3- El-Sayed A, Refaai M, Ateya A. Doppler ultrasonographic scan, gene expression and serum profile of immune, APPs and antioxidant markers in Egyptian buffalo-cows with clinical endometritis. Sci Rep. 2024 Mar 8;14(1):5698. doi: 10.1038/s41598-024-56258-0. PMID: 38459095; PMCID: PMC10923904.
  • 4- Essa B, Al-Sharif M, Abdo M, Fericean L, Ateya A. New Insights on Nucleotide Sequence Variants and mRNA Levels of Candidate Genes Assessing Resistance/Susceptibility to Mastitis in Holstein and Montbéliarde Dairy Cows. Vet Sci. 2023 Jan 3;10(1):35. doi: 10.3390/vetsci10010035. PMID: 36669036; PMCID: PMC9861242.
  • 5- Ateya A, Al-Sharif M, Abdo M, Fericean L, Essa B. Individual Genomic Loci and mRNA Levels of Immune Biomarkers Associated with Pneumonia Susceptibility in Baladi Goats. Vet Sci. 2023 Mar 1;10(3):185. doi: 10.3390/vetsci10030185. PMID: 36977224; PMCID: PMC10051579.
  • Comment: Why was the research material exclusively blood?
  • Response: Buffalos are large animals and it is difficult to get biopsy.
  • The writers greatly value the reviewer's perceptive opinions.
    All the metabolic and hormonal indicators in the examined animals were analyzed in the blood sample. The selection for the investigated genes also could be attributed that they could be expressed in blood
  • Endometritis is associated with systematic reaction that could affect blood; so the explored changes could be judged in blood.

  • Comment: Why in line 207 where the authors write about their experiment the sheep is marked as an experimental model?

Response:  We apologize for this miswriting type. It is corrected.

  • Comment: 6. In Table No. 2, a separate column should be designated for the unit.

Response:  The reviewer's insightful thoughts are much appreciated by the writers. We have included a distinct column for the units of the hormonal and metabolic biomarkers, per the reviewer's recommendations.

  • Comment: 7. I would suggest changing all charts to have one color for the bars.

Response:  We thank the reviewer for his effort. Actually making different bars for each group (healthy and endometritis) is indicative to be easily distinguished. Previous reports made this option as follows:

  • 1- Darwish A, Ebissy E, Ateya A, El-Sayed A. Single nucleotide polymorphisms, gene expression and serum profile of immune and antioxidant markers associated with postpartum disorders susceptibility in Barki sheep. Anim Biotechnol. 2023 Apr;34(2):327-339. doi: 10.1080/10495398.2021.1964984. Epub 2021 Aug 18. PMID: 34406916.
  • 2- Al-Sharif M, Marghani BH, Ateya A. DNA polymorphisms and expression profile of immune and antioxidant genes as biomarkers for reproductive disorders tolerance/susceptibility in Baladi goat. Anim Biotechnol. 2023 Dec;34(7):2219-2230. doi: 10.1080/10495398.2022.2082975. Epub 2022 Jun 7. PMID: 35671246.
  • 3- El-Sayed A, Refaai M, Ateya A. Doppler ultrasonographic scan, gene expression and serum profile of immune, APPs and antioxidant markers in Egyptian buffalo-cows with clinical endometritis. Sci Rep. 2024 Mar 8;14(1):5698. doi: 10.1038/s41598-024-56258-0. PMID: 38459095; PMCID: PMC10923904.
  • 4- Essa B, Al-Sharif M, Abdo M, Fericean L, Ateya A. New Insights on Nucleotide Sequence Variants and mRNA Levels of Candidate Genes Assessing Resistance/Susceptibility to Mastitis in Holstein and Montbéliarde Dairy Cows. Vet Sci. 2023 Jan 3;10(1):35. doi: 10.3390/vetsci10010035. PMID: 36669036; PMCID: PMC9861242.
  • 5- Ateya A, Al-Sharif M, Abdo M, Fericean L, Essa B. Individual Genomic Loci and mRNA Levels of Immune Biomarkers Associated with Pneumonia Susceptibility in Baladi Goats. Vet Sci. 2023 Mar 1;10(3):185. doi: 10.3390/vetsci10030185. PMID: 36977224; PMCID: PMC10051579.

Reviewer 4 Report

Comments and Suggestions for Authors

The authors aimed to investigate the gene expression and serum profile of indicators linked to clinical endo-metritis susceptibility in Egyptian buffalo cows. They examined many factors in relation to metabolic and hormonal biomarkers, etc. Their findings present a viable way to reduce Egyptian buffalo cow endometritis by selectively breeding animals according to genetic markers linked to innate immunity to infection.

The main results are from the RT-PCR. Also, they detected the metabolic and hormonal biomarkers. There's just too little data to draw any conclusion. The authors need to provide more data to support the conclusion.

Comments on the Quality of English Language

The authors aimed to investigate the gene expression and serum profile of indicators linked to clinical endo-metritis susceptibility in Egyptian buffalo cows. They examined many factors in relation to metabolic and hormonal biomarkers, etc. Their findings present a viable way to reduce Egyptian buffalo cow endometritis by selectively breeding animals according to genetic markers linked to innate immunity to infection.

The main results are from the RT-PCR. Also, they detected the metabolic and hormonal biomarkers. There's just too little data to draw any conclusion. The authors need to provide more data to support the conclusion.

Author Response

Reviewer 4

We would like to express our deep gratitude to the reviewer for spending the time necessary to modify our submission.

The authors aimed to investigate the gene expression and serum profile of indicators linked to clinical endo-metritis susceptibility in Egyptian buffalo cows. They examined many factors in relation to metabolic and hormonal biomarkers, etc. Their findings present a viable way to reduce Egyptian buffalo cow endometritis by selectively breeding animals according to genetic markers linked to innate immunity to infection.

  • Comment: The main results are from the RT-PCR. Also, they detected the metabolic and hormonal biomarkers. There's just too little data to draw any conclusion. The authors need to provide more data to support the conclusion.
  • Response: We have made some changes in the conclusion to give better meaning.
  • We have some highlights for our study that promote its acceptance and publications;
  • Our study links between gene expression and biochemical profile of immune, antioxidant and metabolic markers.
  • The role of investigated of investigated genes are reported here for first time in healthy and endometritis buffalo.
  • In our design we have the hypothesis that comparing between gene expression profile and biochemical data is enough to judge the health status of animal (change on the level of gene does not necessitate lea to change in the biochemical profile.
  • Our hypothesis that PCR detects presence of any change even if it is present in trace amount.
  • Changes in the investigated markers associated with endometritis make an obvious standing for these markers and the course of disease. Therefore a justifiable treatment could be done and the best time for treatment could be accomplished.

Round 2

Reviewer 1 Report

Comments and Suggestions for Authors

The authors improved the quality of the manuscript, and clarified most of the comments.

But they forgot to insert most of the response in the manuscript.

Please double check.

L58-60 Reduced pregnancy at first AI, pregnancy rate, and increased days open, prevalence of anovular cows etc.

DOI: 10.1111/rda.14301

L74-77 Cite new information related to ketoprofen (NSAID) as an alternative treatment for uterine diseases.

https://doi.org/10.3168/jds.2023-24585

L78-70   Comment from the round 1 (the sentence was not deleted) Exclude this sentence, those methods are not required, and the reference is related with equine.

L106-110 Insert the number of days in milk or days postpartum when you started the synchronization.

 Please insert your response in the manuscript.

             Comment: MM: Sample size calculation is missing, how many days in milk (DIM) were the cows evaluated, insert the range or interval mean ± standard deviation and number of cows enrolled at 21 days after calving and 26 after calving. When after calving were the synchronized, for example after 45 or 50 DIM.

Response:  The writers greatly value the reviewer's perceptive observations. The sample size used in this study was determined using the sample size determination formula as follows:

Z1-α/2 = Standard normal variant at 5% type I error (P < 0.05); P = expected prevalence based on previous study (Hassan et al., 2020); d = absolute error or precision (which is 5%).

The buffalo cow was assessed at 28–33 DIM. Forty buffalo cows were enrolled at 21 days and 26 days following calving, respectively. Following calving, the buffalo cows were synchronized at 45–50 DIM.

·         Comment: How many cows were evaluated to meet 40 per group? How did you select the 40 control (healthy) cows?bised sample

Response:  The reviewer's astute observations are much valued by the authors. 160 buffalo cows were assessed, with 40 in each group due to the time-consuming and costly nature of laboratory biochemical analysis and gene expression patterns of the immune and antioxidant transcript levels. The control (healthy) buffalo cows were in good health and had normal calving and postpartum conditions (i.e., customary feed consumption, body temperature, and no uterine discharge), while the endometritis group was characterized by purulent or muco-purulent uterine discharge with offensive odor, anorexia, and depression.

Comment: How did you check the vaginal discharge? Metricheck, palpation?

Response:  The authors greatly appreciate the reviewer's insightful observations. We used palpation to check the vaginal discharge.

Please replace Clinical endometritis by purulent vaginal discharge

Double check your response for the random and fixed effects in your model.

Fixed could be your groups for example PVD vs control and the biochemical parameters and immunological could be the response or dependent variables. Please insert your response in the manuscript.

How many days postpartum were blood samples taken?

Please insert the reference value for each variable presented so that the readers could better understand your results.

Please comment that the cholesterol could be used as a negative acute phase protein.

“Cholesterol is a component of lipoproteins, forming lipid composition of cell membranes, its measurement helps assess liver functionality, and cholesterol is considered a negative

acute-phase reactant.” DOI: 10.1111/rda.13768

Author Response

The authors improved the quality of the manuscript, and clarified most of the comments. But they forgot to insert most of the response in the manuscript.

We would like to express our deep thanks and appreciation to the reviewer for the time given to revise our submission.

  • Comment: L58-60 Reduced pregnancy at first AI, pregnancy rate, and increased days open, prevalence of anovular cows etc.

DOI: 10.1111/rda.14301.

Response: The reviewer's informative comments are much appreciated by the authors. We have followed the reviewer's recommendations and included the most recent references regarding the financial losses caused by clinical endometritis and how it affects buffalo cows.

  • Comment: L74-77 Cite new information related to ketoprofen (NSAID) as an alternative treatment for uterine diseases. https://doi.org/10.3168/jds.2023-24585

Response: The writers greatly value the reviewer's perceptive observations. We have complied with the reviewer's suggestions and included fresh data regarding ketoprofen (NSAID) as a substitute therapy for uterine disorders.

  • Comment: L78-70   Comment from the round 1 (the sentence was not deleted) Exclude this sentence, those methods are not required, and the reference is related with equine.

Response:  The writers greatly value the reviewer's perceptive observations. As per the reviewer's suggestions, we have excluded this sentence.

  • Comment: L106-110 Insert the number of days in milk or days postpartum when you started the synchronization.

Response:  The reviewer's insightful views are much appreciated by the writers. We have added the number of days in milk when we began the synchronization in accordance with the reviewer's recommendations

Comment: Please insert your response in the manuscript.

  • Comment: MM: Sample size calculation is missing, how many days in milk (DIM) were the cows evaluated, insert the range or interval mean ± standard deviation and number of cows enrolled at 21 days after calving and 26 after calving. When after calving were the synchronized, for example after 45 or 50 DIM.
  • How many cows were evaluated to meet 40 per group? How did you select the 40 control (healthy) cows?
  • How did you check the vaginal discharge? Metricheck, palpation?
  • Double check your response for the random and fixed effects in your model. Fixed could be your groups for example PVD vs control and the biochemical parameters and immunological could be the response or dependent variables. Please insert your response in the manuscript.
  • How many days postpartum were blood samples taken?

Response:  The reviewer's astute observations are much valued by the authors. In compliance with the reviewer's suggestions, we have included our answers in the manuscript.

  • Comment: Please replace Clinical endometritis by purulent vagina discharge.

Response:  The reviewer offered several very good observations, which the writers really value. Purulent vaginal discharge has been used in place of clinical endometritis per the reviewer's recommendations, and we've included the most recent references below.

  • Comment: MM: Please insert the reference value for each variable presented so that the readers could better understand your results.

Response:  The writers greatly appreciate the insightful points made by the reviewer. Following the reviewer's recommendations, we have added the reference interval in table 2 for every variable that was given.

  • . Comment: Please comment that the cholesterol could be used as a negative acute phase protein. “Cholesterol is a component of lipoproteins, forming lipid composition of cell membranes, its measurement helps assess liver functionality, and cholesterol is considered a negative acute-phase reactant.” DOI: 10.1111/rda.13768

Response:  The reviewer's astute observations are much appreciated by the authors. We have added a new reference to the paper and indicated that the cholesterol might be utilized as a negative acute phase protein in accordance with the reviewer's recommendations

Reviewer 2 Report

Comments and Suggestions for Authors

Thank you for replying to my comments.

I understand your opinion and the way of experimental design.

However, why don't you include a detailed description of your response to the comments in the paper? I thought that if this is not adequately described within the manuscript, the paper has no value. Please ponder this point. If this point is not resolved, I think it is not good.

Comments on the Quality of English Language

I think that the quality of the English in this paper is fair.

Author Response

We wish to thank the reviewer from the bottom of our hearts for taking the time to edit our work.

  • Comment: Thank you for replying to my comments. I understand your opinion and the way of experimental design. However, why don't you include a detailed description of your response to the comments in the paper? I thought that if this is not adequately described within the manuscript, the paper has no value. Please ponder this point. If this point is not resolved, I think it is not good.

Response:  The authors much appreciate the reviewer's insightful observations. As per the reviewer's recommendations, our responses are included to the text.

Reviewer 4 Report

Comments and Suggestions for Authors

NO more comments

Comments on the Quality of English Language

NO more comments

Author Response

Comments and Suggestions for Authors

NO more comments

Comments on the Quality of English Language

NO more comments

Response

We thank reviewer for this good response.

Round 3

Reviewer 2 Report

Comments and Suggestions for Authors

Thank you for selecting me as a reviewer of this manuscript.

Please confirm my comments.

Comments on the Quality of English Language

Thank you for your correction and addition to my comment. I think this is correct. Thank you very much.